# Advancing interactive systems with liquid crystal network-based adaptive electronics

Pengrong Lyu[1,2], Dirk J. Broer ®[1,2] & Danqing Liu[1,2] ✉

Achieving adaptive behavior in artificial systems, analogous to living organisms, has been a long-standing goal in electronics and materials science. Efforts to integrate adaptive capabilities into synthetic electronics traditionally involved a typical architecture comprising of sensors, an external controller, and actuators constructed from multiple materials. However, challenges arise when attempting to unite these three components into a single entity capable of independently coping with dynamic environments. Here, we unveil an adaptive electronic unit based on a liquid crystal polymer that seamlessly incorporates sensing, signal processing, and actuating functionalities. The polymer forms a film that undergoes anisotropic deformations when exposed to a minor heat pulse generated by human touch. We integrate this property into an electric circuit to facilitate switching. We showcase the concept by creating an interactive system that features distributed information processing including feedback loops and enabling cascading signal transmission across multiple adaptive units. This system responds progressively, in a multi-layered cascade to a dynamic change in its environment. The incorporation of adaptive capabilities into a single piece of responsive material holds immense potential for expediting progress in next-generation flexible electronics, soft robotics, and swarm intelligence.

Adaptation, a complex yet essential characteristic in living systems, empowers organisms across various levels to independently process and respond to environmental stimuli through closed-loop feedback mechanisms, relying on sensing, signal processing, and actuating functionality[1,2]. This ability enables their survival within their ever-changing environments. A noteworthy example illustrating adapt-ability in nature is *Mimosa pudica Linn*[2,3]. This plant's leaves and leaflets sequentially fold (see Fig. 1a, left, and Supplementary video 1) when they are exposed to specific environmental cues like touch, temperature variations, or light. This dynamic response enhances its effectiveness in deterring herbivores from startling potential threats[4]. It necessitates three fundamental functions within a single leaf: sensing environmental signals, processing received data, and initiating physical responses. These functions serve as the foundation for creating closed-loop feedback within adaptive units[5–7]. In the realm of artificial

electronics, significant efforts have been directed toward implementing adaptive capabilities[8,9]. These efforts have predominantly focused on developing centralized control systems that utilize additional controllers to interface with separate sensing and actuation modules[10–15]. However, these components, produced from different materials and processing techniques, are typically manufactured independently. This results in material property disparities and variations in signal strength, which hampers the efficient integration of multiple functions into a cohesive entity. Additionally, centralized information processing encounters limitations when dealing with the vast amounts of data transmission and computational demands among different modules in dynamic environments[16,17]. Developing an electronics unit that cooperatively achieves sensing, signal processing, and actuation functions akin to biological systems remains a highly sought-after yet elusive objective[18].

[1]Institute for Complex Molecular Systems, Eindhoven University of Technology, Den Dolech 2, 5612 AZ Eindhoven, The Netherlands. [2]Department of Chemical Engineering and Chemistry, Eindhoven University of Technology, Den Dolech 2, 5612 AZ Eindhoven, The Netherlands. ✉e-mail: danqing.liu@tue.nl

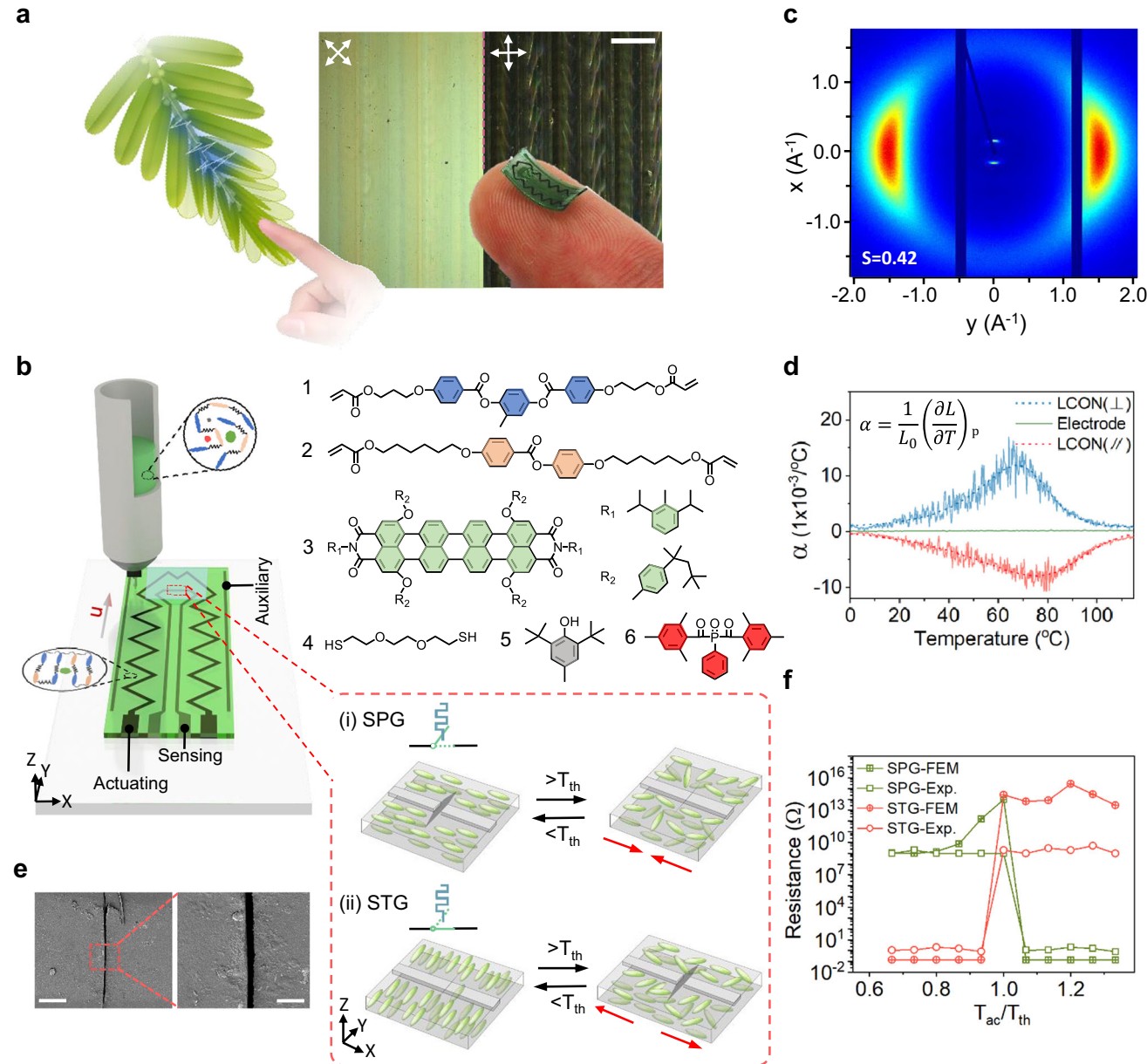

**Fig. 1 | The design strategy of adaptive electronic units. a** Left, schematic of the response of *Mimosa pudica Linn's* leaves upon human touch. Right, a printed adaptive electronic unit mimicking a *Mimosa* leaflet is demonstrated on a human finger. The background image shows the birefringence color of the printed LCON film under the polarized optical microscope. Scale bar, 300 μm. **b** 3D illustration presents the direct ink writing method to fabricate an LCON adaptive unit. Enlarged images elaborate the principle of (i) self-propelled gate (SPG) by the closure of the gap in the conductor during LCON heating and (ii) a self-terminated gate (STG) by interrupting the conductor by heating. Red arrows mark the direction of internal stress. $T_{th}$ denotes the threshold temperature. **c** 2D-WAXD measurement of an LCON at room temperature, suggesting a high scalar order parameter ($S$) in the oligomeric network. **d** Measurement of thermal expansion coefficient ($\alpha$) as a function of temperature. The anisotropic thermal expansion $\alpha$ of LCON is measured in the direction perpendicular to the director ($\perp$) as indicated by the blue line, and the parallel direction (//) is given in the red line. The green line is the $\alpha$ of the electrode. **e** Scanning electron microscopy (SEM) measurements of a micro-gap of about 6 μm created in the SPG at room temperature. Scale bars indicate 100 and 30 μm in the left and right images, respectively. **f** Resistance change as the function of relative temperature $T_{ac}/T_{th}$ in (°C/°C). $T_{ac}$ represents the actual temperature in the LCON. The green line denotes SPG, while the red line indicates STG. Both finite element modeling (FEM) results and experimental results are represented by solid dots and hollow dots, respectively.

Inspired by the macroscopic response of the *Mimosa*, we propose an interactive system that can transfer information based on intrinsic closed-loop feedback and cascade signal transmission. This interactive system consists of multiple adaptive electronics units that integrate sensing, signal processing, and actuating functionalities into a single material (Fig. 1a, right). This material converts the received stimuli, such as heat, light, and touch, into a binary electrical signal, enabling adaptivity to dynamic environmental changes. For this purpose, we design the configuration of our adaptive electronic unit based on

liquid crystal oligomer networks (LCONs), self-assembled anisotropic polymer networks that undergo order-to-disorder transitions in response to external stimuli[19–41]. The designed configuration contains four fundamental components, as shown in the left panel of Fig. 1b. The LCON serves as the core element of the adaptive unit, facilitating both sensing and actuating functionalities. Complementing this, we incorporate two main functional electrodes into the design. The sensing electrode is centrally located within the LCON film, enabling the transmission of electric signals when environmental changes induce

deformations in the LCON. The actuating electrode converts electric signals to thermal signals (Joule heating) for actuation. This component is situated at the LCON periphery and is usually connected in series with the sensing electrode to an external electricity source. This input serves only energy source without electronic control. Additionally, an auxiliary electrode has been included to enhance mechanical stability during actuation without connecting to the circuit. More details about the dimension of the adaptive unit and the connection of the electrodes can be seen in Supplementary Fig. 1.

## Results

The LCON is based on an acrylate-terminated main chain oligomer synthesized via a Michael addition reaction[42–45]. Its composition involves six distinct molecules (Fig. 1b, right), with Molecule 1 serving as the fundamental matrix. Molecule 2 is introduced to lower the phase transition temperature due to its weak π–π interactions and adjusts the phase morphology (Supplementary Fig. 2). Molecule 3 functions as an IR dye for photo-thermal conversion (Supplementary Fig. 3). Molecule 4 serves as the chain extender in the Michael addition reaction. Molecule 5 is employed as an inhibitor to prevent premature curing, while Molecule 6 serves as a photo-initiator with absorption properties in the 300–450 nm wavelength range. A balanced formulation is developed to tune its rheology at an oligomeric state for processing. Moreover, the mixture is adjusted to ensure that the designed thermal strain of our LCON is within a healthy adult finger temperature range between 20 and 37 °C[46]. The fabrication of LCON film is carried out using the direct ink writing method (Fig. 1b, left), where the oligomer mixture is extruded from the nozzle in its nematic state at 30 °C. This process relies on shear forces and elongational flow to align the LC oligomer chains, enabling molecular order parallel to the flow direction[47,48]. The established order can be quantified by the scalar order parameter, which describes the orientational degree of the molecular long axes to a specified direction. A higher value of the scalar order parameter indicates a higher degree of molecular alignment within the LC oligomer. The photo-polymerization is used to crosslink the uniaxial-aligned oligomer into a network. This uniaxial alignment of the LCON is confirmed by polarized optical microscopy (Fig. 1a, right) and wide-angle X-ray diffraction (WAXD) analysis (Fig. 1c). Typically, its diffraction pattern exhibits a smectic layer structure and a nematic structure with an order parameter of ~0.42 at room temperature, which indicates a rather high order in the oligomeric network system[49].

The aligned LCON is capable of detecting environmental changes such as heat, light, and electricity. Upon detection of environment changes, actuation occurs as the LCON undergoes anisotropic geometric deformations, defined by the thermal expansion coefficient. LCON experiences expansion perpendicular to the long axes of the molecular direction and contraction parallel to it (Fig. 1d). The thermal expansion coefficient of the electrode, on the other hand, is positive and has an absolute value 20 times less than our LCON in the 20–40 °C temperature range (Supplementary Fig. 4). Employing these characteristics, we design two types of adaptable logic switches, which is embedded in the sensing electrode. In the first case, the actuation of LCON initiates the closing of the initial open circuit (Fig. 1b(i)). This is achieved by printing the sensing electrode in parallel with the molecular direction of our LCON and creating a micro-gap at its center. The detailed fabrication process can be obtained from Supplementary Fig. 5. A typical gap of about 6 μm is measured by Scanning Electronic Microscopy (SEM), as shown in Fig. 1e. When actuated, this gap narrows and eventually closes once the actual temperature ($T_{ac}$) of the LCON exceeds the threshold temperature ($T_{th}$), which allows the current to pass through the sensing electrode. This process is confirmed by both the finite element modeling (FEM) and the experimental characterizations, as shown in Fig. 1f. More details are shown in Supplementary Fig. 6, Supplementary Videos 2 and 3. This adaptive

switching behavior is an analogy to the binary logic function, which can switch from OFF-state to ON-state upon activation. Subsequently, the LCON device converts the electric signal into a thermal signal, which manifests as Joule heating. This activates the bending deformation of the LCON away from the electrode side. Bending behavior originates from the contrast between the negative thermal expansion coefficient of the uniaxial-aligned LCON and the positive coefficient in the electrode. We investigate this adaptive bending behavior within a real-world dynamic environment by the approach of a human finger. The corresponding equivalent circuit is shown in Fig. 2a. When the finger is in proximity (Fig. 2c), the LCON device experiences temperature changes depending on its distance from the finger. With $T_{ac} < T_{th}$, the LCON device bends slightly to a varying degree and returns to its initial state once the finger is withdrawn (Fig. 2d and Supplementary Fig. 7). There is small hysteresis in the bending curve due to the limited cooling time, which can be eliminated by prolonging the time. At a finger proximity of <1 mm, temperatures above $T_{th}$ are reached, inducing the sensing electrode to close. Consequently, the circuit current is engaged and peaks at 6.2 mA, generating extra Joule heating to amplify its bending deformation even if the finger is retracted. Eventually, this LCON device bends to a stable angle upon reaching thermal equilibrium with the surrounding air. This process establishes a positive feedback loop, consisting of sensing heat from the environment to deform LCON, closing the electric circuit to allow current flow, the current flow inducing Joule heating, and the extra heat increasing the LCON deformation. The entire adaptive bending behavior is shown in Supplementary Video 4. This adaptable logic switch is further referred to as the self-propelled gate (SPG).

In contrast, the self-terminated gate (STG) switches from a closed circuit to an open circuit when- the LCON is actuated, comparable to the binary logic function that switches from the ON-state to the OFF-state upon activation. This is achieved by orienting the sensing electrode perpendicular to the molecular direction of the LCON (Fig. 1b(ii)). To characterize the behavior of the STG, we monitor the voltage across the sensing electrode and the corresponding bending angle of the LCON device. The equivalent circuit is depicted in Fig. 2b. We apply 2.0 V to STG, yet the voltage temporarily stays at 0.25 V due to the in-series connection with the actuating electrode (Fig. 2e). This induces Joule heating and subsequently causes $T_{ac}$ to continue rising (Supplementary Fig. 8). Meanwhile, LCON starts to bend away from the electrode side. When $T_{ac} \geq T_{th}$, the expansion of the LCON is sufficient to cause the micro-gap to open, interrupting the current flow and causing the electric potential of the sensing electrode to equal the 2 V supplied by the power source. As a result, the LCON cools down due to the absence of Joule heating. When $T_{ac}$ falls below $T_{th}$, the sensing electrode closes again and initiates a new heating-cooling cycle. Fundamentally, this process is a negative feedback loop that reaches a self-oscillating state determined by the equilibrium between Joule heating and natural convection cooling. In our typical sample, the initial oscillation frequency is 2 Hz (Fig. 2f), which can be adjusted on-demand by altering power input (Supplementary Fig. 9). When a disturbance, such as direct heating, cooling, or IR irradiation, enters this negative feedback loop, STG oscillates to adapt to the new environmental conditions by autonomously adjusting the Joule heating period, as shown in Fig. 2f (i-iii). After removing the disturbance, STG restores the initial oscillation within 2 s (Supplementary Fig. 8). This entire process is recorded in Supplementary Video 5.

Next, we cascade signals to the second, third, and subsequent LCON adaptive electronic units. To do this, we connect the sensing electrode of the prior unit with the actuating electrode of the succeeding unit, forming a series connection to an electrical source (Fig. 3a). When electricity is applied to the actuating electrode of the prior adaptive unit, the Joule heating causes the deformation of the LCON. Typically, the LCON is heated to 28 °C within 5 s with the input electric power at 50 mW (Fig. 3b). When $T_{ac}$ reaches $T_{th}$, electricity

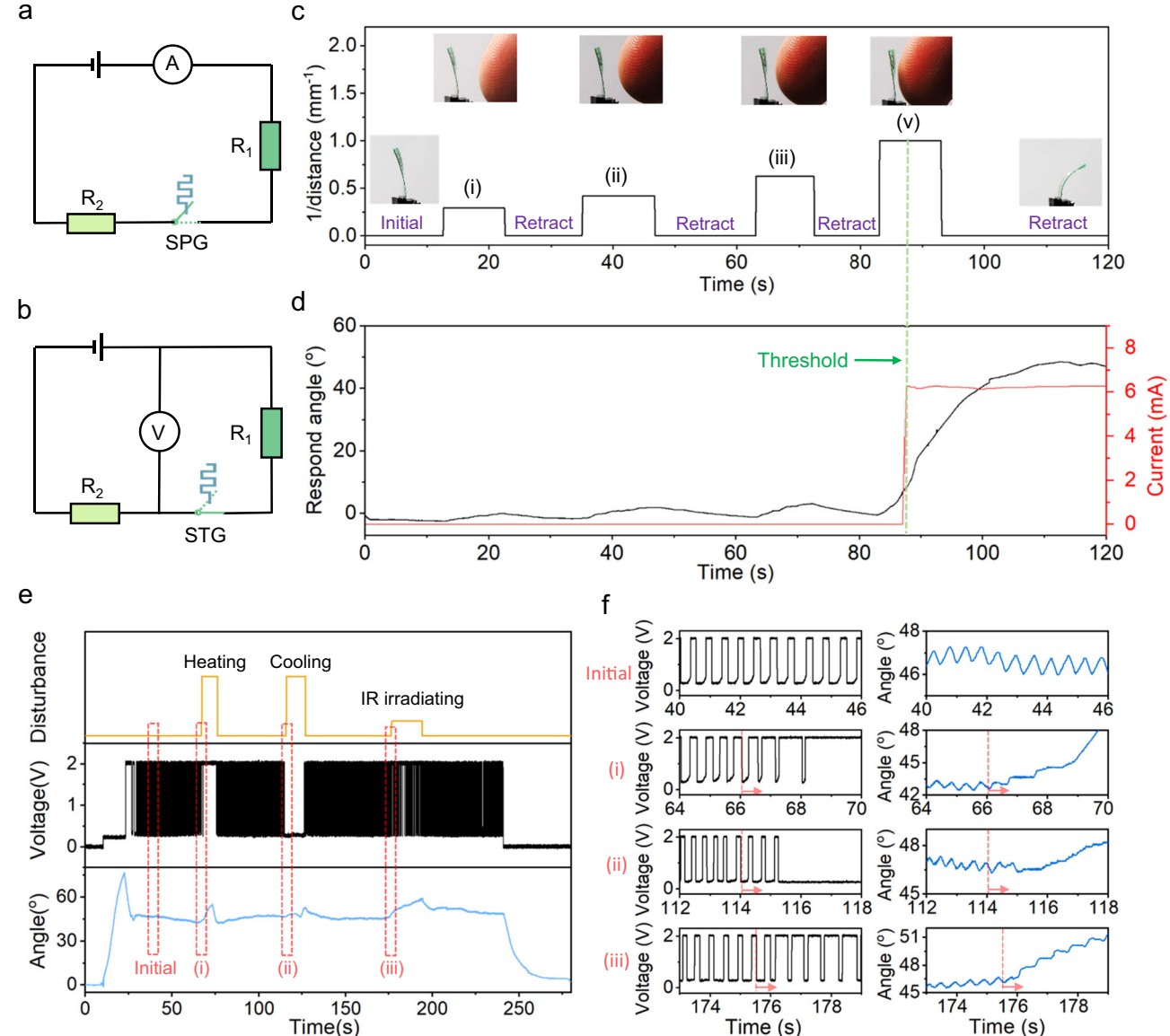

**Fig. 2 | Constructing a closed-loop feedback in a single LCON adaptive electronic unit. a, b** The equivalent electrical circuit of (**a**) a positive feedback loop and (**b**) a negative feedback loop. $R_1$ and $R_2$ correspond to the resistance in the sensing electrode and actuating electrode, respectively. **c, d** The response behavior of SPG when a human finger approaches. **c** A human finger cyclically approaches and withdraws from SPG at varying distances between LCON and finger. The top row are real-life images of the experiment and the bottom plots indicate the

reciprocal of distance between the finger and SPG. **d** Triggered SPG response in bending angle (black curve) and the current flow (red curve). **e** The response behavior of the STG when subjected to different disturbances: (i) direct heating, (ii) cooling by ice, (iii) IR irradiating. **f** The enlarged plots correspond to regions "initial" to (iii) in Fig. 2e. The red arrows indicate that the disturbances are starting to enter the negative feedback loop.

flows from the sensing electrode of the prior SPG to the actuating electrode of the succeeding SPG, causing both units to bend (Fig. 3d). The bending angle of LCON depends on the input voltage (Fig. 3c). During bending, current flows through the sensing electrode of the succeeding SPG (Fig. 3d(iv)). This signal cascade transmission can be further extended by connecting more SPGs in sequence and eventually stopped by connecting an STG, as shown in Fig. 3e, where the output current in the sensing electrode drops from 0.7 to 0 mA. If the first adaptive unit is an STG, no signal can be transmitted (Supplementary Fig. 10).

By modulating the input electric signal and the properties of the LCON, we adjust the signal transmission behavior. Firstly, we investigate the signal transmission rate, which depends on the circuit switching kinetics. This speed is primarily governed by the rate of Joule heating influenced by the input voltage amplitude. To analyze this, we

apply a square wave signal to the actuating electrode and record the corresponding resistance change on the sensing electrode (Fig. 3f(i) and g(i)), with the delay between them defining circuit kinetics. Increasing the input voltage leads to a higher transmission rate in SPG (Fig. 3f(ii)) and a termination rate in STG (Fig. 3g(ii)). Secondly, we investigate the threshold temperature to activate the LCON logic switch. When the temperature is below $T_{th}$, LCON deformation is insufficient to switch the circuit. The deformation required to switch the circuit is initialized by the micro-gap processing. When LCON cools down to room temperature, the width of the previously formed gap changes. Typically, the micro-gap forms directly after crosslinking above room temperature. In principle, possible gap widths are defined by the cross-linking temperatures, which can range from below room temperature to close to the nematic-isotropic transition temperature of the oligomer. A higher cross-linking temperature leads to a larger

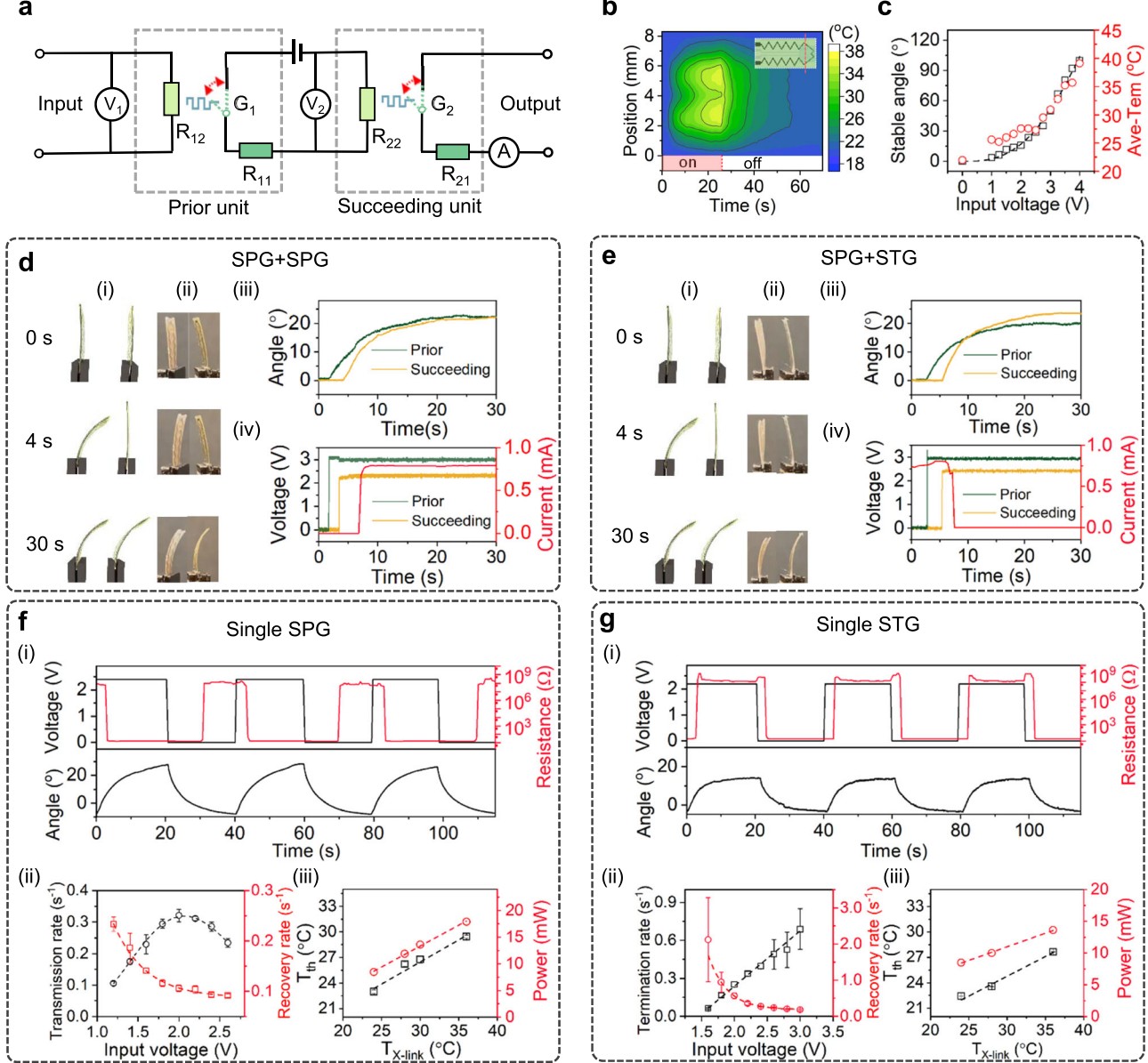

**Fig. 3 | Signal cascade transmission between multiple adaptive electronic units.**
**a** The equivalent electrical circuit of signal cascade transmission in two adaptive electronics. The $R_{11}$ and $R_{21}$ represent the resistance of the sensing electrode in the prior and succeeding units, respectively. $R_{12}$ and $R_{22}$ indicate the resistance of the actuating electrode in the prior and succeeding units, respectively. $G_1$ and $G_2$ denote the adaptable logic switch in the prior and succeeding units, respectively. **b** Temperature increases of the LCON over time with 50 mW input power. The red line in the inset indicates the position of measurement. **c** Stable bending angle and the corresponding average temperature increase with increasing input voltage. **d** Signal cascade transmission between two SPGs, and **e** signal cascade transmission from SPG to STG. (i) The schematics of their deformation. (ii, iii) The corresponding experimental results. (iv) Voltage and current changes during signal cascade. **f** Signal transmission in a single SPG, and **g** signal termination in a single STG. For both cases, the input is a square wave signal. (i) The output resistance change is recorded, also the change in the bending angle is measured. (ii) The influence of input voltage on the transmission rate and termination rate (black curve), quantified by the reciprocal of transition time from high resistance to low resistance and vice versa, respectively. The recovery rates are quantified by the the reciprocal of required time for their opposite process (red curve). The error bars indicate the standard deviation of the transmission rate and termination rate over 5 tests. (iii) The influence of cross-linking temperature on the threshold temperature (black curve) and the driven power (red curve).

change in gap width, consequently increasing the $T_{th}$ and the required input power (Fig. 3f(iii) and g(iii)). In SPG, the gap widens, while in STG, it narrows and eventually closes (Supplementary Fig. 6). This actuation is dependent on the resistance of the actuating electrode, where a higher resistance requires a higher input threshold voltage, as shown in Supplementary Fig. 11. The deactivation of the LCON logic switch is affected by the $T_{ac}$ induced by the applied voltage, with higher voltage resulting in a lower recovery rate (Fig. 3f(ii) and g(ii)).

Based on the newly developed knowledge of adaptive electronics, we create an artificial *Mimosa* by coupling the positive feedback loop and signal cascade transmission. To replicate the touch-sensitive capabilities of a real *Mimosa* (Fig. 4a(i)), we carefully choose a specific cross-linking temperature for the LCON to match the temperature of the examiner's finger. In this case, derived from Fig. 3f(iii), we select a cross-linking temperature of 26 °C, corresponding to a threshold temperature of 24 °C. For illustration, we connect eight SPGs on a flexible printed circuit board (PCB), as shown in Fig. 4a(ii). The detailed design specifications and circuit are provided in Supplementary Fig. 12. The simplified schematic of signal flow and equivalent circuit of the artificial *Mimosa* are

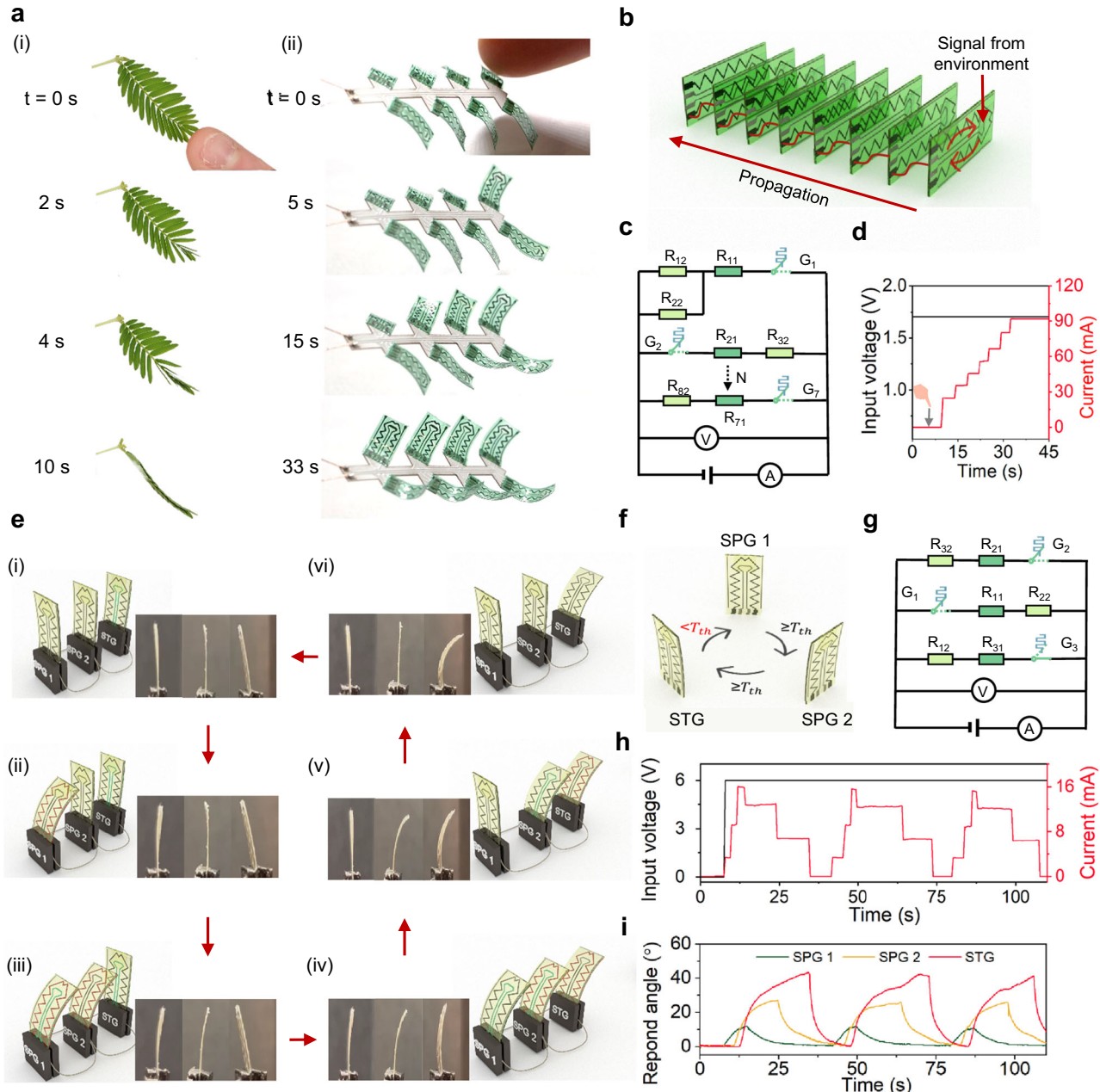

**Fig. 4 | Assembling an interactive system. a** Mimicking a *Mimosa* using SPGs. (i) Photographs of a real *Mimosa* whose leaves fold sequentially upon human touch. (ii) Artificial *Mimosa* that constitutes of eight SPGs. **b** Schematics of signal flow in simplified Fig. 4a(ii). **c** The equivalent circuit of (**b**). **d** Current and voltage changes during the actuation of the artificial *Mimosa*. The gray arrow marks the examiner's finger touching an artificial *Mimosa*. **e** A collective oscillating system consists of two SPGs and one STG. From (i) to (vi), both experimental results and schematics are given. Sensing electrodes in green indicate that the circuit is closed, while gray indicates an open state. Actuating electrodes in red indicate the activation of Joule heating. **f** Simplified schematics of signal flow of (**e**). **g** The equivalent circuit of **e**. The $R_{11}$, $R_{12}$, and $G_1$ represent the resistor in the sensing electrode, the resistor in the actuating electrode, and the logic switch of the SPG1, respectively. The $R_{21}$, $R_{22}$, $G_2$, and $R_{31}$, $R_{32}$, and $G_3$, represent the corresponding resistors and logic switches in the SPG2 and STG, respectively. **h** Input voltage and current change of the system over time. **i** The bending angle of each adaptive unit during oscillation.

presented in Fig. 4b and c, respectively. The first SPG leaf of the artificial *Mimosa* forms a positive feedback loop by connecting its sensing electrode and actuating electrode in series to a constant voltage source, initiating the signal transmission with the environment. This sensing electrode also connects to the actuating electrode of subsequent SPG, enabling the signal cascade transmission. When the examiner's finger makes contact with the first SPG leaf, it initiates a slight bending motion, which is further amplified by the positive feedback loop. This amplification allows for the transmission of the electric signal, even if the finger is withdrawn. As shown in

Fig. 4d, the current in the artificial *Mimosa* increases step-wise, indicating that the eight SPG leaves are activated sequentially. Hereby, we successfully emulate a real *Mimosa*, which acts upon human touch with a sequential folding response (Supplementary video 6).

So far, we have created a positive feedback loop in the first SPG to initiate signal cascade transmission in the system through multiple SPGs. To further expand the capabilities of our system, we incorporate a negative-feedback-loop-inducing STG as the final unit to terminate signal transmission such that creates a collective self-oscillating

system (Fig. 4e). To achieve this, we connect two SPGs and an STG in series to a constant voltage source, where the output from the sensing electrode of the STG serves as the input signal for the actuating electrode of the first SPG. The schematic of the signal flow and the equivalent circuit are shown in Fig. 4f and g, respectively. Upon applying voltage, the first SPG bends, activated by Joule heating due to the closed circuit of the STG (Fig. 4e(ii), h, and i). This bending behavior is propagated to the second SPG and STG (Fig. 4e(iii, iv)). During this process, the temperature within the STG gradually rises. Once it reaches the predefined $T_{th}$, the STG interrupts the current flow to the first SPG, restoring the system to its initial state (Fig. 4h, i). Subsequently, the temperature decreases until it falls below the threshold, upon which the STG re-establishes the current flow through the first SPG and initiates a new heating–cooling cycle. This cyclic process describes a collective self-oscillation within the system, as shown in Supplementary Video 7.

## Discussion

Based on a liquid crystal oligomer network, we have developed an interactive system containing multiple adaptive units that can independently realize a closed-loop of "sense-decide-act". We apply this behavior to facilitate dynamic interactions between units or between units and their surroundings. By engineering molecular orientation, either parallel or perpendicular to the sensing electrode, two binary electronic logic operations are demonstrated, namely self-propelled gate (SPG) and self-terminated gate (STG). SPG initiates signal transmission and establishes a positive feedback loop, while STG stops signal transmission and enforces a negative feedback loop. The advantages of our adaptive unit are: that it is an intrinsic sensor and actuator, therefore, it only requires energy demand ≤20 mW. Furthermore, the signal transmission speed in our system is in the time frame of seconds, which is comparable with the real Mimosa. To further illustrate the versatility of our adaptive unit, we created an artificial *Mimosa* that exhibits a sequential folding response upon human touch by coupling closed-loop feedback and cascading transmission mechanisms across multiple SPGs.

Our hybrid conductive-LCON presents a new advance in flexible electronics, where the electrodes are embedded in a responsive substrate. It further creates opportunities for next-generation soft robotics, allowing them to adapt to unpredictable natural environments. Moreover, the closed-loop characteristic in our LCON device enables mutual communications between units independently, which provides a capability to address the computational challenges in swarm systems with numerous entities in dynamic environments. We anticipate that our adaptive electronics unit will inspire new design possibilities for integrating memory functions, advancing toward self-learning intelligent systems and swarm intelligence.

## Methods
### Materials
An overview of the materials for the adaptive electronics unit is provided in Fig.1b (right). Molecule **1** 2-Methyl-1,4-phenylene bis (4-(3-(acryloyloxy)propoxy)benzoate) was purchased from Merck. Molecule **2** 4-(6-(Acryloyloxy)hexyloxy)phenyl 4-(6-(acryloyloxy)hexyloxy) benzoate was acquired from Daken Chemical Limited. Molecule **3** Lumogen IR 788 was procured from BASF. Molecule **4** 2,2'-(Ethylenedioxy) diethanethiol, Molecule **5** Butylated hydroxytoluene and basic catalyst (1,8-diazabicyclo[5.4.0]undec-7-ene, DBU) were obtained from Sigma-Aldrich. Molecule **6** (Irgacure 819) was purchased from Ciba. All molecules were used as received from the supplier. Polyvinyl alcohol (PVA) with an average molecule weight of 25,000 g/mol was obtained from Sigma-Aldrich. dichloromethane (DCM) was obtained from Biosolve. Two conducting inks, PE671 (carbon-based) and PE874 (silver-based) were purchased from Dupont. Another silver-based conductive ink (Elepaste NP1) was acquired from Taiyo.

### Synthesis of liquid crystal oligomer
Liquid crystal oligomers were synthesized using a base-catalyzed thiol-acrylate Michael addition. Typically, the molar ratio of acrylate to thiol is approximately 1:0.91, and the molar ratio of Molecule **1** to Molecule **2** is 3:2. To optimize the thermal properties of the LCON, the ratio of Molecule **1** varied. When making the adaptive units for investigating signal cascade transmission and collective oscillation, Molecule **3** was omitted. The standard synthesis procedure involved combining Molecule **1** (1412.6 mg), Molecule **2** (861.7 mg), Molecule **3** (15 mg), Molecule **4** (663.57 mg), Molecule **5** (2.5 mg), and Molecule **6** (20 mg) in a vial. Subsequently, 10 mL of DCM was added to dissolve the mixture, followed by the addition of the base catalyst DBU (10 μL). After stirring for 3 h at room temperature and an additional 3 h at 50 °C to evaporate part of the DCM, the solution was transferred to a poly(tetrafluoroethylene) evaporation dish. It was then placed in a vacuum oven at 45 °C overnight to remove the remaining DCM solvent. Finally, the oligomer was loaded into an ink syringe for 4-D extrusion printing.

### Preparation of the conductive inks
Conductive ink for printing actuating electrodes was prepared by blending the PE671 and PE874. The two conductive inks were directly added to the vial and mechanically stirred for 10 min. The conductivity of the bending mixture increased with higher PE874 concentrations (Supplementary Fig. 13). Typically, a mixture containing 50 wt% PE874 and 50 wt% PE671 was used for preparing the actuation electrode. When fabricating the adaptive unit for the collective oscillation system, a mixture containing 40 wt% PE874 and 60 wt% PE671 was used. For printing the sensing and auxiliary electrodes, a pure commercial conductive ink (Elepaste NP1) was used. Its conductivity is similar to that of pure PE874. Finally, the two tailored conductive inks were loaded into separate syringes.

### The fabrication of the adaptive electronic unit
Using commercial 3D printers (EHR, Hyrel 3D), a multi-step direct ink writing procedure (Supplementary Note 1 and Supplementary Fig. 5) was used to manufacture the adaptive electronics unit. To ease the removal of adaptive units after printing, a PVA sacrificial layer was spin-coated onto the glass substrate before printing. Following this, the PE671 and PE874 conductive ink blend and the pure Elepaste NP1 were used to print the actuating electrode and sensing and auxiliary electrodes, respectively. Pre-designed printing paths were used, after which the conductive inks were heat-cured. Above the conductive ink layers, the liquid crystal oligomer was extruded onto the substrate at the area of the logic switch (blue boxed area in Fig. 1b (left) and photopolymerized above room temperature. Additional oligomer material was then extruded onto the substrate in the regions surrounding the rest area. Unlike the curing process for the primary printed oligomer area, the secondary printed area was typically photopolymerized at ambient temperature. Next, the printed samples were carefully peeled off their substrate after immersion in water for 3 h to dissolve the PVA layer. The freed samples were then dried in air to remove any water remaining on their surfaces. Finally, the sensing electrodes and the LCON film were cut through with a handmade die, thus creating the initial micro-gap in the logic switch region. This process was performed above room temperature, enabling the initial micro-gap to close or open further upon cooling, depending on the alignment of the liquid molecules within the LCON.

### Differential scanning calorimetry (DSC)
DSC measurements were employed to determine the transition temperature of oligomers in TA Instruments DSC Q1000. The sample was heated and cooled at a rate of 10 °C min⁻¹ from −50 to 100 °C. This cycle was performed twice, with the first cycle being used to eliminate the thermal history accumulated during the preparation of the sample.

## X-ray diffraction analysis (XRD)

X-ray scattering measurements were performed on a Ganesha lab instrument equipped with a Genix-Cu ultralow divergence source that generates X-ray photons with a wavelength and flux of 0.154 nm and $1 \times 10^8$ photons s$^{-1}$, respectively. Diffraction patterns were obtained using a Pilatus 300 K silicon pixel detector with $487 \times 619$ pixels of $172 \times 172\ \mu m^2$. Silver behenate was used as a calibration standard. The sample-to-detector distance was 89 mm for wide-angle (WAXS) configurations, which allowed for LCON molecular morphology and order parameters to be determined.

## Dynamic mechanical thermal analysis (DMTA)

Using a TA Instruments Q800 device in vertical tension mode, dynamic mechanical thermal analysis was performed on thin films. The force-controlled mode was chosen to measure the thermal strain in the LCON film with uniaxial alignment. The sample was cooled to 0 °C and equilibrium was held for 5 min with a pre-load force of 0.005 N. The 0.005 N static force was retained while the sample was heated to 120 °C and cooled at a rate of 5 °C/min. This cycle was performed twice to remove the thermal history of the material. Young's modulus in the aligned LCON with varying orientations and the cured conductive ink was measured by cutting the film into rectangular pieces about 5 mm by 15 mm in size. The film had a thickness of ~100 μm. Thermographs were obtained with a single 1 Hz oscillating frequency, 0.05 N preload force, 50 μm amplitude, and between −20 and 100 °C at a heating rate of 5 °C min$^{-1}$.

## Electrical characterization

To measure resistance and current changes during sensing and actuating, the LCON device was mounted on a 3D-printed custom-built clamp where four copper wires with a diameter of 100 μm were fixed on the surface. Two copper wires were connected to the sensing electrodes, while the other two were connected to the actuating electrodes. To provide a constant input voltage for the LCON device, a DC power source (TENMA 72-2720) was used. The resistance and the current of the LCON device were measured using a source meter (Keithley 2400), and a digital oscilloscope (RTB 2004) was used to track the voltage. Matlab (R2019a) was used as a control platform during the measurement.

## Optical characterization

The alignment of the printed LCON film was checked using a polarization optical microscope (Leica DM 2700M). The absorption of IR dye was measured using a Perkin Elmer Lambda 750 UV/Vis/NIR spectrophotometer. A digital camera (OLYMPUS E-M10 Mark IV, 60 mm lens) was used to capture photographs and videos. A USB camera (60 fps, 2 MP) was used to investigate the closed feedback loop in the single adaptive electronic unit. An infrared camera (Xenics, Gobi +640 GigE) was used to capture thermal images/videos. Video analysis software (Tracker, version 6.1.0) was used to track the bending deformation of the adaptive electronics unit.

## SEM imaging

A JEOL JSM-IT100 scanning electron microscope operating at a 5 kV acceleration voltage was used to capture the SEM images to study the micro-gap evolution in samples.

## Simulation analysis

All FEM analysis was conducted using the COMSOL Multiphysics (v.6.1a). The detailed setup and parameters can be found in Supplementary Note 2 and Supplementary Fig. 14.

## Data availability

The data that support the findings of this study are available from the corresponding author upon request. Source data are provided with this paper.

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

## Acknowledgements

We thank the financial support from the ICMS-CRT STA Neural Networks project. We thank Shiyi Xia and León David Nijenhuis for helping us in building the electronic setup for the measurements. We thank Sam Weima for designing the PCB for the artificial *Mimosa*, Dr. Sean Lugger for XRD measurement, and Tom Bus for SEM measurement. We thank Charlotte Kjellander from Holst Centre for her help in the conductive inks. We thank Mert Orhan Astam for his help in editing.

## Author contributions

D.J.B. and D.L. conceived the research and supervised the project. P.L. performed experiments, analyzed the data, and drafted the manuscript. P.L., D.J.B. and D.L. wrote the manuscript.

## Competing interests

The authors declare no competing interests.
