## [Peer Review File · Nature Communications]

REVIEWER COMMENTS

Reviewer #1 (Remarks to the Author):

Comments #1: The manuscript reports the systematic study of a new interactive self-adaptation elastomer device, which combines heat sensing and actuation of liquid crystal elastomer material with multiple adaptive units, which inspires the development of soft robotic elastomer materials in practical applications, allowing them to adapt to unpredictable natural environments. The author 3D printed the all-in-one elastomer device with commercialization capability and describes detailed usage of two novel adaptable logical switch SPG & STG by engineering liquid crystal molecular orientation to realize a closed-loop of “sense-decide-act” interacting between units or between units and their surroundings, bio mimicking the Mimosa leaf. The results are very interesting and may receive considerable attention from the general readers. The research work is well organized, and the data is systematically logical, having both experimental measurements and simulated results well explained. It is acceptable for publication after minor revisions.

Comments #2: In line 61-64 states: “However, these components, produced from different materials and processing techniques, are typically manufactured independently. This results in material property disparities and variations in signal strength, which hampers the efficient integration of multiple functions into a cohesive entity.”, to point out independent materials and processing techniques are drawbacks of current adapting methods. But next sentences in line 67 “Developing an electronics unit that independently achieves sensing, signal processing, and actuation functions, akin to biological systems, remains a highly sought-after yet elusive objective.” also using word “independently” here conflicts with line 61-64. The authors may want to change a word opposite to “independently”, may be using “cooperatively” here to emphasizes that developing a new interactive self-adaptation system is a highly sought-after yet elusive objective. Also, a reference (Engineering, 7(5), 581-602.) related to developing soft electronics with sensing, adaptation and actuation functions is recommended to be cited here.

Comment #3: In Figure 2 d,e, f, the heat bending angle curve has hysteresis or heat history (more obviously in supplementary figure 5 and extended data figure 1 f), please explain and how to use this device in practical device applications like soft Robotics if with hysteresis or heating history.

Comment #4: The equivalent circuit figure 4c needs more detailed circuit for each SPG, please add the full circuit drawing on the supplementary fig 8. In Figure 4g, please write down which resistor R_{xx} and G_x numbers correspond to STG, SPG1 or SPG2 in Figure 4e for better understanding.

Reviewer #2 (Remarks to the Author):

In this manuscript, Lyu and coworkers have fabricated an interactive system based on LCEs, that shows information processing, feedback loops and adaptive response to dynamic stimulation. Creating adaptability and feedback loops in materials is a longstanding challenge and is highly sought after in small-scale and soft robotics. This work showcases how integrating a thermo/photo-responsive material into internal soft circuitry can make the creation of dynamic behaviour with feedback loops achievable.

After reading the manuscript a couple of times I could not find any major points that should be addressed or improved. This manuscript is close to perfect, very well-written, well-presented, and timely. As such, I think it can be published in Nat. Comm. as is.

Reviewer #3 (Remarks to the Author):

The paper, "Advancing Interactive Systems with Liquid Crystal Network-Based Adaptive Electronics" by P. Lyu, et al., presents a printed liquid crystal elastomer (LCE) actuator integrated with soft electrodes. This innovative design incorporates dual functionality, serving as both a Joule heat actuator and a mechanical electrode gate within a single LCE piece, facilitating positive and negative feedback. Various units are interconnected to create intriguing actuator systems. Mimosa mimicry and collective oscillators are demonstrated within the study.

In my opinion, the term 'adaptation' has often been misused in the literature on stimuli-responsive materials. In this manuscript, the authors present a new model system that effectively embodies the bio-inspired concept of adaptation.

The authors employed a clever technique by creating micro-gaps in the soft electrode, enabling actuation-to-connect and actuation-to-close responsiveness. These two responses serve as positive and negative feedback in their circuit. It is noteworthy that positive feedback is rarely reported in contemporary literature.

Positive feedback induces bistable stages, while negative feedback facilitates self-oscillation. The authors have effectively demonstrated their findings with comprehensive characterization.

In an extension of this concept, the micro-gap can control the opening and closing of another actuator through wire connection. This feedback mechanism can be applied to other elements, enabling closed-loop and network realization. The conceptual level of this work clearly exceeds the average of publications in Nature Communications, in my opinion.

Hopefully my comments can contribute to further enhancing the quality of the manuscript

1. Line 39-40, "a range of stimuli", but it did not follow by a range of stimuli.
2. Line 45, "single artificial electronic units" to "single piece of responsive material", or something like that, I would suggest.
3. (Major) Line 49 in introduction and else. "Adaptation" is a complex term in biology encompassing various concepts such as homeostasis (negative feedback loops), positive feedback, growth, reproduction, and more. Research conducted on Mimosa often associates adaptation with the learning process to train the plant to respond to complex stimuli. Therefore, I believe that Mimosa is not an ideal example to illustrate adaptation in line with the research presented in this manuscript. What the authors have observed seems more akin to dynamic or cascading responses. In my view, the key concepts highlighted in this manuscript are closed feedback loops and cascading signal transmission.

But, the way of illustrating the concept, is fully up to the authors.
4. Line 73 "enabling full adaptivity", cannot be "full".
5. line 83, "external electricity source to receive electrical input." I suggest to add, input serves only energy source, with no electronic control.
6. (Major) I found Fig. 1b, "sensing" electrode misleading. I guess a "gate" may help better understanding. While talking about feedback, readers are expecting to know where the loop is. There are four electrodes in Fig.1b, connection is yet known. Reading until Fig.2a,b, readers probably still don't understand how the electrodes are connected. Readers may find it challenging to grasp the concept.
7. (Major) I suggest moving the description of the 'positive feedback loop' mentioned in line 130 to the end of the paragraph. Readers should first be introduced to your circuit, the entire closed loop, then to grasp the concept of positive feedback. Additionally, in my opinion, the description of positive feedback is either inaccurate or unclear. Positive feedback typically involves A inducing B, B inducing C, and then C enhancing A. In this case, I would propose that the current induces Joule heat, heat induces LCE deformation, LCE deformation closes the gap, and the gap closure enhances the current.
8. Fig.2, "varying distances"? I guess it is the distance between LCE and finger?
9. Fig. 4 i, the angles should not be normalized, since each actuator is different in function, i.e., STG or SPG.
10. Line 252 on. It is rather unusual to compare power consumption with that of other model systems, particularly in the conclusion section. Heat actuation is greatly influenced by size (heat

capacity), so direct comparisons with different systems based solely on their power values may not be appropriate. Additionally, this manuscript introduces a novel concept rather than addressing a practical aspect.

11. Line 255 on. Mimosa is a complex plant that reacts to touch in a sequence, suggesting the presence of cascading signal transitions. This manuscript vividly presents a biomimetic model. Additionally, Mimosa can return to its original state, indicating the presence of homeostasis. References 49-50 explore homeostasis mechanisms driven solely by negative feedback. It is inappropriate to compare the response speeds between the systems based on hydrogels.

12. In Supplementary Figure 6, panel (b), what level of input power was utilized? The meaning the error bars?

-H.Z.

Reviewer#1

Reviewer #1 (Remarks to the Author):

Comments #1: The manuscript reports the systematic study of a new interactive self-adaptation elastomer device, which combines heat sensing and actuation of liquid crystal elastomer material with multiple adaptive units, which inspires the development of soft robotic elastomer materials in practical applications, allowing them to adapt to unpredictable natural environments. The author 3D printed the all-in-one elastomer device with commercialization capability and describes detailed usage of two novel adaptable logical switch SPG & STG by engineering liquid crystal molecular orientation to realize a closed-loop of “sense-decide-act” interacting between units or between units and their surroundings, bio mimicking the Mimosa leaf. The results are very interesting and may receive considerable attention from the general readers. The research work is well organized, and the data is systematically logical, having both experimental measurements and simulated results well explained. It is acceptable for publication after minor revisions.

Reply: We thank the reviewer for the positive comment.

Comments #2: In line 61-64 states: “However, these components, produced from different materials and processing techniques, are typically manufactured independently. This results in material property disparities and variations in signal strength, which hampers the efficient integration of multiple functions into a cohesive entity.”, to point out independent materials and processing techniques are drawbacks of current adapting methods. But next sentences in line 67 “Developing an electronics unit that independently achieves sensing, signal processing, and actuation functions, akin to biological systems, remains a highly sought-after yet elusive objective.” also using word “independently” here conflicts with line 61-64. The authors may want to change a word opposite to “independently”, maybe using “cooperatively” here to emphasize that developing a new interactive self-adaptation system is a highly sought-after yet elusive objective. Also, a reference (Engineering, 7(5), 581-602.) related to developing soft electronics with sensing, adaptation and actuation functions is recommended to be cited here.

Reply: We thank the kind comments and valuable insights from the reviewer. In the revised manuscript, we have replaced the word “independently” with “cooperatively” in line 68. And the recommended reference (Engineering, 7(5), 581-602) has been cited in line 69.

Comment #3: In Figure 2 d,e, f, the heat bending angle curve has hysteresis or heat history (more obviously in supplementary figure 5 and extended data figure 1 f), please explain and how to use this device in practical device applications like soft Robotics if with hysteresis or heating history.

Reply: Indeed, the reviewer is correct that there is a small hysteresis in our materials. Regarding Figures 2d, e, and f, the hysteresis of the bending angle in our device is relatively small, less than 2° in comparison with the activated angle $> 40^\circ$. To eliminate this hysteresis, we can increase the cooling rate which can be controlled by changing the environmental conditions, e.g. using an active cool element. Alternatively, we can prolong the cooling time.

In the Figure 2d, the cooling time is 10 seconds. However, in the other figures of the paper, e.g. Figure 3f, g, and extended data Figure 4a,b, the cooling time is 20 seconds, we can clearly see that hysteresis is eliminated. We added a sentence “There are small hysteresis in the bending curve due to the limited cooling time, which can be eliminated by prolonging the time” in the main text in lines 139-140 to make it clearer. We also explained this information in the caption of Supplementary Figure 5.

In the extended data Figure 1, the hysteresis in the thermal strain curve originated from material properties. This is unlike the previous discussion about Figure 2d,e, and f on the bending angle which is caused by the device properties. In this case, the increased cooling rate will enlarge the hysteresis while the slowed cooling rate will reduce the hysteresis. This is because the transition kinetics of LC mesogens from a disordered to an ordered state is controlled by the cooling rate. We have performed additional experiments in Supplementary Figure 11, in which we have both increased and decreased the cooling rate. We note that the hysteresis can be hypothetically eliminated if the thermal rate is infinitely small. In the revised extended data, we add this information in the caption of the extended data Figure 1.

The remark of the reviewer on the application of the thermal hysteresis effect inspires us to come up with some new ideas. Firstly, we can think of tunable memory material in which the memory period of the material can be even by controlling the thermal rate. Secondly, we can consider the naturally triggered stent with delayed deployment functionality for medical applications.

Comment #4: The equivalent circuit figure 4c needs more detailed circuit for each SPG, please add the full circuit drawing on the supplementary fig 8. In Figure 4g, please write down which resistor R_{xx} and G_x numbers correspond to STG, SPG1 or SPG2 in Figure 4e for better understanding.

Reply: Adding more details in the circuit indeed helps the better understanding. We have included the full circuit drawing and written down the resistor R_{xx} and G_x numbers corresponding to eight SPGs in the supplementary Figure 8. In addition, we have written down all the resistor R_{xx} and G_x numbers corresponding to SPG1, or SPG2 and STG in the caption of Figure 4 in the revised manuscript.

Reviewer#2

Reviewer #2 (Remarks to the Author):

In this manuscript, Lyu and coworkers have fabricated an interactive system based on LCEs, that shows information processing, feedback loops and adaptive response to dynamic stimulation. Creating adaptability and feedback loops in materials is a longstanding challenge and is highly sought after in small-scale and soft robotics. This work showcases how integrating a thermo/photo-responsive material into internal soft circuitry can make the creation of dynamic behaviour with feedback loops achievable. After reading the manuscript a couple of times I could not find any major points that should be addressed or improved. This manuscript is close to perfect, very well-written, well-presented, and timely. As such, I think it can be published in Nat. Comm. as is.

Reply: We express our sincere gratitude to the reviewer for the kind comments.

Reviewer#3

The paper, "Advancing Interactive Systems with Liquid Crystal Network-Based Adaptive Electronics" by P. Lyu, et al., presents a printed liquid crystal elastomer (LCE) actuator integrated with soft electrodes. This innovative design incorporates dual functionality, serving as both a Joule heat actuator and a mechanical electrode gate within a single LCE piece, facilitating positive and negative feedback. Various units are interconnected to create intriguing actuator systems. Mimosa mimicry and collective oscillators are demonstrated within the study.

In my opinion, the term 'adaptation' has often been misused in the literature on stimuli-responsive materials. In this manuscript, the authors present a new model system that effectively embodies the bio-inspired concept of adaptation.

The authors employed a clever technique by creating micro-gaps in the soft electrode, enabling actuation-to-connect and actuation-to-close responsiveness. These two responses serve as positive and negative feedback in their circuit. It is noteworthy that positive feedback is rarely reported in contemporary literature.

Positive feedback induces bistable stages, while negative feedback facilitates self-oscillation. The authors have effectively demonstrated their findings with comprehensive characterization.

In an extension of this concept, the micro-gap can control the opening and closing of another actuator through wire connection. This feedback mechanism can be applied to other elements, enabling closed-loop and network realization. The conceptual level of this work clearly exceeds the average of publications in Nature Communications, in my opinion.

Hopefully my comments can contribute to further enhancing the quality of the manuscript.

Reply: We thank the reviewer for providing thoughtful comments to further improve our manuscript.

1. Line 39-40, "a range of stimuli", but it did not follow by a range of stimuli.

Reply: We have removed "a range of stimuli" from the sentence in lines 38-40.

2. Line 45, “single artificial electronic units” to “single piece of responsive material”, or something like that, I would suggest.

Reply: We have replaced “single artificial electronic units” with “single piece of responsive material” in lines 44-45.

3. (Major) Line 49 in introduction and else. “Adaptation” is a complex term in biology encompassing various concepts such as homeostasis (negative feedback loops), positive feedback, growth, reproduction, and more. Research conducted on Mimosa often associates adaptation with the learning process to train the plant to respond to complex stimuli. Therefore, I believe that Mimosa is not an ideal example to illustrate adaptation in line with the research presented in this manuscript. What the authors have observed seems more akin to dynamic or cascading responses. In my view, the key concepts highlighted in this manuscript are closed feedback loops and cascading signal transmission. But, the way of illustrating the concept, is fully up to the authors.

Reply: Indeed, “adaptation” is a complex characteristic in living systems. The reviewer has pointed out in his previous comments that “In my opinion, the term “adaptation“ has often been misused in the literature on stimuli-responsive materials. In this manuscript, the authors present a new model system that effectively embodies the bio-inspired concept of adaptation.” We agree with this remark that in our manuscript “adaptation” is appropriate. We would like to clarify the word “adaptation” as the achievement of adaptation often requires closed feedback loop characterization, including sensing, signal processing, and actuating functions. In our work, we demonstrate how the sensing, signal processing, and actuating functions can be integrated into a single piece of responsive LCON material, allowing the material to adaptively respond to the dynamic environment. We have further included this information and better defined the word “adaptation” in lines 49-51 of the main text for clarification.

Mimosa is known for its touch-induced leaf-folding behavior, relying on the sensing, processing, and actuating functions in its single leaflet. Many researches have been performed in this respect, e.g. DOI: 10.1039/c7tc04879c, DOI: 10.1016/j.matt.2021.11.031, DOI: 10.1002/sml.201403036. Therefore, from this point of view, we believe that our single LCON device is very similar to the response behavior of a single Mimosa leaflet. Furthermore, the Mimosa leaves fold sequentially upon a human touch the first leaflet as shown in Supplementary Video 1. The mechanical folding in the first leaflet can be propagated to the next leaflet. This sequential folding process is exactly what we mimic in our system. Therefore, we believe that Mimosa is a suitable analogous example for the reader to easier understand our work. Nevertheless, we would like to emphasize that we use Mimosa as an analog for its macroscopic behavior and not to focus on its biological mechanism. We have further clarity on this point in line 70 of the main text.

4. Line 73 “enabling full adaptivity”, cannot be “full”.

Reply: We have deleted the “full” from the sentence in line 74.

5. line 83, “external electricity source to receive electrical input.” I suggest to add, input serves only energy source, with no electronic control.

Reply: We have added one sentence “This input serves only energy source, without electronic control” in line 85.

6. (Major) I found Fig. 1b, “sensing” electrode misleading. I guess a “gate” may help better understanding. While talking about feedback, readers are expecting to know where the loop is. There are four electrodes in Fig.1b, connection is yet known. Reading until Fig.2a,b, readers probably still don’t understand how the electrodes are connected. Readers may find it challenging to grasp the concept.

Reply: We thank the kind suggestions from the reviewer. Actually, we have defined a “logic switch”, similar to the “gate” function as the reviewer mentioned in the main text. The logic switch is embedded into the sensing electrode. To make the sensing electrode and logic switch clearer, we changed the sentence “Employing these characteristics along with the integrated sensing electrode, we design two types of adaptable logic switches connected to an electricity source” to “Employing these characteristics, we design two types of adaptable logic switches, which is embedded in the sensing electrode” in line 119-120.

Additionally, to make the connection of the electrodes clearer, we added a new Supplementary Figure 1h to explain the connections within the electrodes. In the main text, we replaced the original sentence with “This component is situated at the LCON periphery and is usually connected in series with the sensing electrode to an external electricity source. This input serves only energy source, without electronic control. Additionally, an auxiliary electrode has been included to enhance mechanical stability during actuation without connecting to the circuit. More details about the dimension of the adaptive unit and the connection of the electrodes can be obtained from Supplementary Fig. 1.” in lines 83-88.

7. (Major) I suggest moving the description of the 'positive feedback loop' mentioned in line 130 to the end of the paragraph. Readers should first be introduced to your circuit, the entire closed loop, then to grasp the concept of positive feedback. Additionally, in my opinion, the description of positive feedback is either inaccurate or unclear. Positive feedback typically involves A inducing B, B inducing C, and then C enhancing A. In this case, I would propose that the current induces Joule heat, heat induces LCE deformation, LCE deformation closes the gap, and the gap closure enhances the current.

Reply: We have moved the description for the positive feedback loop to the end of the paragraph. Additionally, we added one sentence “This process establishes a positive feedback loop, consisting of sensing heat from the environment to deform LCON, closing the electric circuit to allow current flow, the current flow inducing Joule heating, and the extra heat increasing the LCON deformation.” in lines 144-147 to make the description of the positive feedback clearer.

8. Fig.2, “varying distances”? I guess it is the distance between LCE and finger?

Reply: The reviewer is correct. We’ve replaced “ varying distances” with “varying distance between LCON and finger” in the caption of Figure 2 to make it clearer.

9. Fig. 4 i, the angles should not be normalized, since each actuator is different in function, i.e., STG or SPG.

Reply: We modified Figure 4i to show the bend angle without normalization. The SPG1, SPG2, and STG exhibit different maximum bend angles. That is because the resistance of the actuating electrode in the SPG1, SPG2, and STG are different during fabricating the sample.

10. Line 252 on. It is rather unusual to compare power consumption with that of other model systems, particularly in the conclusion section. Heat actuation is greatly influenced by size (heat capacity), so direct comparisons with different systems based solely on their power values may not be appropriate. Additionally, this manuscript introduces a novel concept rather than addressing a practical aspect.

Reply: As suggested by the reviewer, we have removed the comparison of the power consumption and deleted the reference.

11. Line 255 on. Mimosa is a complex plant that reacts to touch in a sequence, suggesting the presence of cascading signal transitions. This manuscript vividly presents a biomimetic model. Additionally, Mimosa can return to its original state, indicating the presence of homeostasis. References 49-50 explore homeostasis mechanisms driven solely by negative feedback. It is inappropriate to compare the response speeds between the systems based on hydrogels.

Reply: We have deleted the comparisons with references 49-50 and changed the sentence to “Furthermore, the signal transmission speed in our system is in the time frame of seconds, which is comparable with the real Mimosa” in lines 260-261.

12. In Supplementary Figure 6, panel (b), what level of input power was utilized? The meaning the error bars?

Reply: The input power was 40.12 mW when we investigated the influence of IR irradiation on frequency. The error bar means the standard deviation of the oscillatory frequency over 30s. In the revised Supplementary information, we added the input power and the meaning of the error bar in the caption of Supplementary Figure 6.

REVIEWERS' COMMENTS

Reviewer #1 (Remarks to the Author):

The revised paper is ready for publication. Thanks Lyu and coworkers for detailed replies on all the reviewer's comments and the questions are well addressed and resolved in revised manuscript.

Reviewer #3 (Remarks to the Author):

The authors have addressed my concerns.

There are "bosom friends" from other disciplines, the authors are suggested to add one reference, [Self-Sustained and Coordinated Rhythmic Deformations with SMA for Controller-Free Locomotion. Adv. Intell. Syst. 2024, 2300667.] in which very similar concept was raised, from a mechanical engineering aspect.

Above paper is not reviewer's own publication.

There is no need to return back for another round of review.

Reviewer#1

Reviewer #1 (Remarks to the Author):

The revised paper is ready for publication. Thanks Lyu and coworkers for detailed replies on all the reviewer's comments and the questions are well addressed and resolved in revised manuscript.

Reply: We express our sincere gratitude to the reviewer for the kind comments.

Reviewer #3 (Remarks to the Author):

The authors have addressed my concerns. There are "bosom friends" from other disciplines, the authors are suggested to add one reference, [Self-Sustained and Coordinated Rhythmic Deformations with SMA for Controller-Free Locomotion. Adv. Intell. Syst. 2024, 2300667.] in which very similar concept was raised, from a mechanical engineering aspect. Above paper is not reviewer's own publication. There is no need to return back for another round of review.

Reply: We thank the reviewer for recommending the reference. We added the recommended paper "Adv. Intell. Syst. 2024, 2300667" in line 58.